# Advanced Research on Biological Properties—A Study on the Activity of the *Apis mellifera* Antioxidant System and the Crystallographic and Spectroscopic Properties of *7-Diethylamino-4-hydroxycoumarin*

**DOI:** 10.3390/ijms26147015

**Published:** 2025-07-21

**Authors:** Klaudia Rząd, Iwona Budziak-Wieczorek, Aneta Strachecka, Patrycja Staniszewska, Adam Staniszewski, Anna Gryboś, Alicja Matwijczuk, Bożena Gładyszewska, Karolina Starzak, Anna A. Hoser, Maurycy E. Nowak, Małgorzata Figiel, Sylwia Okoń, Arkadiusz Paweł Matwijczuk

**Affiliations:** 1Department of Biophysics, University of Life Sciences in Lublin, Akademicka 13, 20-950 Lublin, Poland; klaudia.rzad@up.lublin.pl (K.R.); alicja.matwijczuk@up.lublin.pl (A.M.); bozena.gladyszewska@up.lublin.pl (B.G.); 2Department of Chemistry, University of Life Sciences in Lublin, Akademicka 15, 20-950 Lublin, Poland; iwona.budziak@up.lublin.pl; 3Department of Invertebrate Ecophysiology and Experimental Biology, University of Life Sciences in Lublin, Doświadczalna 50a, 20-280 Lublin, Poland; aneta.strachecka@up.lublin.pl (A.S.); patrycja.staniszewska@up.lublin.pl (P.S.); grybosanna156@gmail.com (A.G.); 4Department of Biotechnology, Microbiology and Human Nutrition, Faculty of Food Science and Biotechnology, University of Life Sciences in Lublin; Skromna 8, 20-704 Lublin, Poland; adam.staniszewski@up.lublin.pl; 5Department of Chemical Technology and Environmental Analytics (C1), Faculty of Chemical Engineering and Technology, Cracow University of Technology, Warszawska 24, 31-155 Kraków, Poland; karolina.starzak@pk.edu.pl; 6WCH UW, Faculty of Chemistry, University of Warsaw, Pasteura 1, 02-093 Warsaw, Poland; a.hoser@uw.edu.pl (A.A.H.); me.nowak8@student.uw.edu.pl (M.E.N.); 7Department of Physical Biochemistry, Faculty of Biochemistry, Biophysics and Biotechnology, Jagiellonian University, Gronostajowa 7, 30-387 Kraków, Poland; m.figiel@uj.edu.pl; 8Institute of Plant Genetics, Breeding and Biotechnology, University of Life Sciences in Lublin, Akademicka 15, 20-950 Lublin, Poland; sylwia.okon@up.lublin.pl

**Keywords:** coumarin derivatives, antioxidants, honeybee, X-ray crystallography, molecular spectroscopy

## Abstract

The search for substances that increase the immunity of bees is becoming a necessity in the era of various environmental threats and the declining immunocompetence of these insects. Therefore, we tested the biological and physicochemical properties of *7-diethylamino-4-hydroxycoumarin* (7DOC). In a cage test, two groups of bees were created: a control group fed with sugar syrup and an experimental group fed with sugar syrup with the addition of 7DOC. In each group, the longevity of the bees was determined and the protein concentrations and antioxidant activities in the bees’ hemolymph were determined. The bees fed with 7DOC lived 2.7 times longer than those in the control group. The protein concentrations and activities of SOD, CAT, GPx and GST, as well as the TAC levels, were significantly higher in the hemolymph of the supplemented workers. To confirm these potent biological properties of 7DOC, the UV-Vis spectra, emission and excitation of fluorescence, synchronous spectra and finally the fluorescence lifetimes of this compound were measured using the time-correlated single photon counting method, in various environments differing in polarity and in the environment applied in bee research. This compound was shown to be sensitive to changes in solvent polarity. The spectroscopic assays were complemented with crystallographic tests of the obtained monocrystals of the aforementioned compounds, which attested to the aggregation effects observed in the spectra measurements for the selected coumarin. The research results confirm that this compound has the potential to be implemented in apiary management, which will be our application goal, but further research into apiary conditions is required.

## 1. Introduction

Honeybees are the most important pollinators worldwide, contributing to the production of approximately 70% of crops used for human consumption. These crops comprise fruits, nuts and vegetables as well as oilseeds that provide the bulk of micronutrients in the human diet [1,2,3]. The value of pollinated crops is estimated at EUR 22 billion per year in Europe and EUR 153 billion per year worldwide [4,5]. Pollinator declines, particularly in the populations of western honeybees, are a major societal challenge. In the US, the number of colonies has been reduced by half since 1940, while the acreage of honeybee-dependent crops has increased. In recent years, more than 1 million colonies have been lost per year in the US and in Europe. Dramatic honeybee winter colony losses have been reported frequently in different regions all over the world, especially in the last 15–20 years [6,7]. These colony depopulation processes are thought to be caused by multiple factors, such as high levels of pathogens/parasites [8,9,10], environmental pollutants [9,11], pesticides [9,12,13], destruction of habitat [14], nutritional stress [15,16], inadequate beekeeping management and climate change [17,18,19]. These losses of bee colonies adversely affect ecosystem services (biodiversity), farm economics and, consequently, human health [2,20,21]. That is why many research centers around the world are searching for ways to increase the immunity of bees. Possible solutions include biostimulators, phytochemicals and/or plant extracts [22]. Positive effects on bees have been proven for caffeine, coenzyme Q10, curcumin, piperine [23], CBD oil [24], hemp extract [25], protoporphyrin [26,27], vitamin C [28], lactic acid [29], resveratrol [30,31], essential oils of mint, melissa, coriander and thyme [32], clove oil, gallic acid, kaempferol and p-coumaric acid [33,34] and even propolis [35,36].

In this research, we decided to check the properties of a compound from the coumarin group, and this choice was dictated by its structural similarities to the compounds mentioned earlier. Coumarins are heterocyclic, oxygen-containing compounds consisting of a combined pyrone and benzene molecule, initially isolated from various plants in the 1820s by Vogel and Guibourt [37]. Coumarins are widely present in almost every part of the plant, such as the bark, seeds, fruits, roots and leaves. However, the extraction yield of coumarins is limited due to the small amounts of them that are present in plants [38]. Therefore, synthetic methods of obtaining coumarins are sought. The extraction efficiency of coumarins in extracts depends on the extraction techniques and the extracted plant material. Furthermore, modern isolation techniques, such as countercurrent chromatography, allow for the extraction of coumarins in sufficient quantities and with sufficient purity for various biological assays. The main structure of coumarins has also been chemically modified to explore coumarins’ pharmacological and biological applications. Modifications of the basic coumarin ring, such as the thiadiazole–coumarin hybrids presented in our previous studies, showed strong antifungal or antibacterial activity [39,40]. Many derivatives of the above-mentioned compounds, owing to their very attractive photophysical properties [41], are also outstanding molecular sensors used for enzymatic tests [42], laser dyes or excellent ligands forming complexes with selected rare earth metal ions [43]. Our other studies show that compounds from this group are also a superb research route to achieving room temperature phosphorescence (RTP) [44]. Its applications include protection against counterfeiting and encryption/decryption of information, optical sensors (temperature, pH, chemical analytes), bioimaging/diagnosis, organic light-emitting diodes, photovoltaic devices or photodynamic therapy, which is a very important complement to oncological treatment. Therefore, after preliminary research, we chose a structurally known compound, *7-diethylamino-4-hydroxycoumarin* (7DOC, Figure 1) and attempted to check the biological properties related to the above-mentioned topics. Its properties were characterized by investigating the antioxidant system activity in bees.

Assuming that strengthening the antioxidant system through dietary supplements improves the vitality and immunity of bees, one of our research objectives was to determine the effect of 7DOC on the activity of the antioxidant system in *Apis mellifera* worker bees during the aging process. Additionally, taking into account the lack of publications strictly collecting the spectral properties of the above-mentioned analogue, we decided to provide a detailed description of the molecular processes occurring in several environments with variable polarity, using methods in the field of optical spectroscopy and X-ray crystallography, including the mixture (sugar syrup) that was used for the bees. We decided to apply molecular spectroscopy methods complemented with measurements of 7DOC monocrystals in order to examine in detail how this compound acts on a molecular level in different solvents, particularly in the bee nutrient solution fed in our study.

## 2. Results and Discussion

### 2.1. Determination of Biological Properties

The bee antioxidant system is an element of its humoral response to environmental pressure, including pathogens [45,46]. Four main enzymes making up the basic first lines of defense against oxidants, i.e., superoxide dismutase (SOD), catalase (CAT), peroxidase (GPx) and glutathione S-transferase (GST), trap oxygen radicals and transform them into less reactive forms [47,48,49]. The oxidative changes and efficiency of the entire antioxidant system in bees are quantified by the total antioxidant capacity (TAC) levels [50,51,52]. TAC results from the interaction between different antioxidants, which provide protection against the effects of ROS, and is regarded as indicating the homeostasis between ROS and antioxidants (identified and unidentified, measurable and non-measurable). Intracellular TAC is mainly attributed to the enzymes, whereas plasma TAC is mainly related to dietary low molecular weight antioxidants [52,53].

In our experiment, worker bees fed with 7DOC lived on average 66 days longer than control bees fed with sugar syrup (1:1) (Figure 1; *p* ≤ 0.01). In total, 50% of the bees in the control group lived to day 18, while 50% of those in the group treated with 7DOC lived to day 38 of the experiment.

The protein concentrations in the hemolymph increased with the age of workers in both groups but were always higher in bees consuming 7DOC in the sugar syrup (Figure 2). A similar tendency was observed for the activities of antioxidants (SOD, CAT, GST and GPx) and levels of TAC, which increased systematically with the age of the workers, regardless of the group, but were always higher in those fed with 7DOC (Figure 3 and Figure 4).

The results are as follows:SOD—superoxide dismutase (two-way ANOVA; group: F_(1,268)_ = 756.20, *p* = 0.0000; days of supplementation: F_(4,264)_ = 5729.4, *p* = 0.0000, se ± 0.033727 (1st day), se ± 0.023849 (7–35 days); group × days of supplementation: F_(4,264)_ = 340.50; *p* = 0.0000); se ± 0.115141 (control), se ± 0.178103 (7DOC);CAT—catalase (two-way ANOVA; group: F_(1,268)_ = 987.15, *p* = 0.0000; se ± 0.129998 (control), se ± 0.280503 (7DOC); days of supplementation: F_(4,264)_ = 2336.2, *p* = 0.0000, se ± 0.069046 (1st day), se ± 0.048823 (7–35 days); group × days of supplementation: F_(4,264)_ = 548.70, *p* = 0.0000, se ± 0.069046);GST—glutathione S-transferase (two-way ANOVA; group: F_(1,268)_ = 639.91, *p* = 0.0000; se ± 0.187744 (control), se ± 0.288596 (7DOC); days of supplementation: F_(4,264)_ = 1948.2, *p* = 0.0000, se ± 0.088550 (1st day), se ± 0.062614 (7–35 days); group × days of supplementation: F_(4,264)_ = 208.68, *p* = 0.0000, se ± 0.088550);GPx—Glutathione peroxidase (two-way ANOVA; group: F_(1,268)_ = 718.09, *p* = 0.0000, se ± 0.168185 (control), se ± 0.331584 (7DOC); days of supplementation: F_(4,264)_ = 1078.7, *p* = 0.0000, se ± 0.105886 (1st day), se ± 0.074873 (7–35 days); group × days of supplementation: F_(4,264)_ = 634.88, *p* = 0.0000, se ± 0.105886).

The simplicity and versatility of the coumarin scaffold make it an interesting starting point for extraction, synthesis and a wide range of applications. Nowadays, many coumarin compounds are examined for use in medicine as compounds with strong pharmacological activities, a broad spectrum, high bioavailability, low toxicity and few side effects, lesser drug resistance, better curative effects, etc. [54,55]. The large-conjugated system in the coumarin ring, with electron-rich and charge-transport properties, is important in the interaction of this scaffold with molecules and ions [56]. Mathivanan et al. [57] suggest that the hydroxyl group in 7DOC facilitates the coordination behavior with metal ions, and the diethylamino group influences the optical features of this molecule. The unification of both functional groups provides interesting features in the molecule in the detection of Al^3+^ and PPi, and in such biochemical reactions as energy transduction, extracellular signal mediations, cyclic adenosine monophosphate synthesis, metabolic processes, etc. It can therefore be assumed that these features make 7DOC a very strong antioxidant, which has a positive effect on bees (Figure 1, Figure 2, Figure 3 and Figure 4). This is supported by the surprisingly long life of these bees, up to 104 days (Figure 1). Naturally, summer bees live for about 3–4 weeks, and winter bees last for about 6–7 months in a temperate climate zone [58]. There is no data in the literature about summer bees surviving for so long in cage tests, and this is a great surprise to the authors of the work. It is likely that 7DOC influences gene expression and most likely changes in the epigenome. However, this requires further research. Even royal jelly (at the concentration of 4%; RJ) in sugar syrup [59] did not extend the life of bee workers as much as 7DOC did. Bernklau et al. [34] and Okińczyc et al. [60] showed that *p*-coumaric acid (from pollen, nectar, honey and bee bread) and coumarins (from propolis) increase worker longevity and pathogen tolerance. Johnson et al. [61] and Mao et al. [62] reported that *p*-coumaric acid and other phenolic acids in honey not only improved longevity but also upregulated detoxifying enzymes. Moreover, Todorov et al. [63] showed that adding a hydroxyl group in position 4 to coumarins improved peroxide scavenging under the experimental conditions. The antioxidant properties of coumarins were also confirmed by Kecel-Gunduz et al. [64], Al-Majedy et al. [65], Bubols et al. [66] and others.

### 2.2. Determination of Crystallographic Properties

#### 2.2.1. Crystal Structure of *7-Diethylamino-4-hydroxycoumarin* (7DOC)

To check the compound’s behavior in the solid state, to shed more light on the ability of the 7DOC molecule to form aggregates and to investigate its potential intermolecular interactions, we crystallized the compound from propan-2-ol. The compound crystallizes in a monoclinic crystal system within the P2_1_/c (14) space group (Table 1). There is only one molecule in the asymmetric unit. 7DOC is a molecule composed of two aromatic rings: the benzene and heterocyclic lactone ring. Molecules in the crystal lattice are L-shaped, with a flat aromatic part and ethyl groups, which are positioned rather perpendicularly to the rest of the compound (Figure 5).

In the 7DOC crystal structure, there are only two types of prevalent intermolecular interactions: *π*-*π* staggered stacking and H-bonds between the hydroxyl and carbonyl groups (Figure 6). The unusual feature of H-Bond O2-H2A … O3 is its rather short distance between the donor and acceptor, 2636(2) Å, while a typical length for such interactions in similar molecules corresponds to a mean distance of 2.83 Å, based on CSD [67] data. Molecules related via hydrogen bonds form infinite flat chains, parallel to the **b** axis, with molecules placed perpendicularly to each other, creating a zigzag pattern. Those chains, made from planar molecules, form layers by weak C-H…*π* interactions. Furthermore, those layers are stabilized with aforementioned *π*-*π* staggered stacking perpendicular to the mean plane of the molecules (Figure 6). Dimer interaction energies for molecules connected via H-Bond O2-H2A … O3 equal −72 kJ/mol and −62 kJ/mol for molecules that interact with *π*-*π* staggered stacking.

The packing of molecules is presented in Figure 7, where we can see a zigzag pattern along the **a** axis, which is typical for aromatic compounds, whereas the parallel chains along **b** are poorly visible, but the layered structure displays aliphatic chains grouped together.

#### 2.2.2. Results of Measurements Made by Means of Absorption Spectroscopy and Stationary Fluorescence with the RLS Technique

However, before a given analogue is brought to the apiculture market, all its properties must be explored thoroughly. For creating a marketable product, it is extremely important to identify the molecular properties of a given compound and how it interacts in a specific environment with variable polarity. This can be achieved, for example, by means of spectroscopic methods, which often remain undefined even for existing analogues, but which enable us to determine the way intermolecular interactions occur. Thus, in a further stage of the study, we undertook to analyze how the chosen derivative, i.e., 7DOC, behaves and what spectral properties it demonstrates in environments with different polarity and in the bee food. Although the structure of this compound is known, so far it has not been analyzed and presented in such a complex manner as in this study.

Figure 8A shows the electron absorption spectra normalized in the absorption peak obtained for 7DOC in several solvents with variable polarity and in the water/sugar syrup. All the values of the exact position of the absorption maxima for the solvents chosen in our study are presented in Table 2. It is evident that as the polarity of the applied solvent increases, an effect consisting of a considerable bathochromic shift (Δλ = 18 nm (~1504 cm^−1^) Figure 8A) of the band maximum from 337 nm to 355 nm appears. Thus, the sensitivity of the compound to a change in the solvent’s polarity is demonstrated distinctly. This is associated with the sites in the structure of this analogue where hydrogen bonds can be quite easily formed with other molecules of the analogue itself and with molecules of the solvent, depending on its type. This was shown in detail in the crystallographic assays, as displayed in Figure 6. For the electron absorption spectra shown in the figure, a broad band from ~310 nm to 370 nm appears, with the maximum in a range from ~330 to ~360 nm linked to the main electron transition S_0_ → S_1_ in the molecule chromophore. As is known from the literature data, in the given case, this maximum is characteristic of the π → π* electron transition [68]. However, it should be noted that an increase in the polarity of the solvent is accompanied by a considerable solvatochromic shift of the maximum of the main band. And another observation was that for most of the spectra obtained on the longwave side, the band is visibly fortified, with the maximum of ~360 nm, e.g., for ethanol. Further, for most of the spectra obtained, we observe a distinct band of much lower intensity with the maximum ~395 nm. Its much lower intensity and the strong bathochromic shift prove that various aggregated forms of a molecule can appear in a given chromophore system, as shown in Figure 8 [69]. This is supported by the results of the crystallographic assays, where we could observe a few structures of potential dimers capable of forming larger, N-aggregating structures.

As suggested by the exciton-splitting theory, it is possible to determine first and foremost what kind of aggregates in a given system appear. For longwave bands, these are card pack arrangements [70]. In addition, we can measure intramolecular distances in an aggregated system for its nearest neighbors, using the following Equation (1):(1)$$R_{\beta }=\sqrt[{3}]{\frac{\mu^{2}\kappa }{\eta^{2}\beta }}$$
where *μ* is the dipole moment of transition of the interacting molecules, *η* is the refractive index and *β* is the dipole–dipole interaction energy (in the classical approach). As proposed in M. Kasha’s exciton-splitting theory, *κ* = 1 for the card pack molecule arrangement in the aggregate and *κ* = −2 for the head-to-tail aggregate [70]. The measured distance between the nearest neighbors in the card pack arrangement was ~3.05 Å and is therefore comparable to the results obtained from the crystallographic tests reported above. This is a distance characteristic of this type of chromophore system [71]. Although, as was emphasized in the analysis of the crystallographic results, 7DOC molecules can occur at quite close distances. The absorption spectrum in the water/sugar syrup, that is in the food given to the bees, lend itself to interesting observations. It was evident that the system was mostly aggregated, the intensity of the spectrum within the range of 330–360 nm decreased considerably, the main peak of absorption on the longwave side broadened and once the main band became stable, it considerably surpasses the other bands in this area. Unquestionably, we are dealing here with the molecular aggregation process in the given environment, which, because of the environment’s properties, seems to be quite a natural development. A preliminary discussion of the tendency for various aggregation effects to occur can be found above, in our description of the crystallographic results. However, it can be assumed that such particle aggregation may promote the absorption of food/syrup containing 7DOC through the bee’s digestive tract into its hemolymph and then into the fat body, where proteins and other compounds, including those included in the antioxidant system, are synthesized [46]. However, further studies need to be conducted on other derivatives of this group of compounds and similar ones, which we are, of course, continuously pursuing. Hence, the high antioxidant activities observed in our studies, shown in Figure 2, Figure 3 and Figure 4.

In the subsequent stage of spectroscopic measurements, considering the usually high quantum efficiencies of compounds from this group, the fluorescence emission spectra were determined. Figure 8B–D show interesting results for the solvents corresponding to those in Figure 8A. Panel B illustrates the fluorescence emission spectra at excitation 335 nm, panel C shows them at excitation 355 nm and panel D shows them at longwave excitation with the maximum at 395 nm. It appears that there is a considerable shift in the peak of the fluorescence emission spectrum depending on the changes in the solvent’s polarity, and this effect is also dependent on the length of the excitation wave. In panel B, depending on the solvent, the emission spectrum shifts from ~380 nm (in EtOH) to ~405 nm for THF. In Panel C, at excitation at 355 nm, the emission spectrum also shifts from ~390 nm to ~405 nm depending on the solvent. As mentioned before, in our description of the absorption spectra, specific bands correspond to specific molecular forms of the compound. Excitation at 335 nm should be linked to the monomeric form, while excitation at 355 nm is to a larger extent associated with the aggregated form, firstly with the dimer type of aggregates, which was demonstrated via crystallographic tests, where we could suspect the presence of a few types of aggregates, of which three seemed most likely to occur. Obviously, there are more possibilities of aggregation effects in a solution, mainly through hydrogen bonds, which is a consequence of a much greater conformational capacity of the environment. As demonstrated in Panels B and C, there are similar peaks of fluorescence emission, which however differ visibly in the intensity of the bands. A change in the intensity of the observed emission spectra is also a proof of the strong aggregation effects in the system submitted to research. A very interesting effect is displayed in Panel D in Figure 8, where the emission spectra are presented for the maximum of the longwave aggregate at 395 nm. Here, we can observe the maximum from ~440 nm (in BtOH) to as much as ~472 nm in the water/sugar solution; this maximum represents a greater bathochromic shift than that detected for water alone. As also seen in Panels B and C, the greater the aggregation process, the greater the extinguishing of the emission maximum, an effect even more evident than in Panels B and C. Thus, the results display emission bands that differ depending on the excitation and the applied medium with the maximum ~400 nm and 440–460 and even further in the water/sugar syrup fed to the bees.

Such large shifts in the emission spectra together with the interpretation of the electron absorption spectra clearly indicate that the fluorescence effects are also related to the process of strong molecular aggregation. Resuming our discussion of the fluorescence effects, it needs to be emphasized that these results confirm the outcome of the spectroscopic assays. The most probable dimer arrangements are head-to-tail conformations in two different energy settings, although other aggregation conformations are also possible, especially in solvent systems.

Thus, the next stage of our research comprised measurements of the fluorescence excitation spectra, and some of the results are illustrated in Figure 9A. In excitation spectra, which are more selective than absorption spectra, it is possible to excite specific spectra forms. Figure 9A shows the excitation spectra for 7DOC in solvents and in the water/sugar syrup, at short- and longwave excitations. As seen in these diagrams, depending on the type of solvent and the length of the wave, the longwave band corresponding to the aggregated forms of the 7DOC molecule is more clearly visualized than in the absorption spectra in Figure 8A.

The aggregation processes occurring in our solvent samples, and especially in the water/sugar solution, are confirmed by the results presented in Figure 9B concerning the measurements of the RLS (resonance light scattering) synchronous spectra. Producing the RLS spectra for 7DOC corresponding to the spectra from Figure 9, and taking into account the research by Pasternack and Collings [72], where the occurrence of RLS spectra is directly linked to the presence of aggregated forms of the analyzed compound (such as dimers or N-aggregates), RLS bands with varied intensity are observed in practically all the solvents tested. However, the biggest RLS bands appear in the solvents, water and water/sugar mixture. These solvents are fundamental solutions in beekeeping. Water is essential to the lives of bees and all other creatures on Earth, and water/sugar solution is used by beekeepers to feed colonies before wintering and during nectarless periods. During these periods, biostimulators, protein products, amino acids or pollen are also fed to bees in syrup or sugar candies [23,24,25,26,27,28,29,30,31,32,33,34]. Undoubtedly, 7DOC can be added to this list. The structure of this compound, based on two main aromatic rings, a benzene ring and a heterocyclic lactone ring, is compatible with naturally occurring compounds (e.g., in plant pollen) whose potential is utilized by bees [73,74,75]. 7DOC, as a coumarin derivative, similarly likely inhibits the proliferation of the bee sacbrood virus (CSBV) in the bee’s body and reduces the number of copies of this virus; at the same time, it may induce the expression of an endogenous antibacterial peptide and improve the innate immunity of bees [76]. These compounds increase the survival of bee larvae [76] and imago, which was also confirmed by our research (Figure 1). Moreover, the oscillatory structure of the RLS bands indicates the presence in the solvent of various associated forms of 7DOC, which has been mentioned a few times earlier in this paper. The presence of various aggregated forms, in turn, affects the position of the fluorescence emission spectra. This likely influences the activity and biological properties of 7DOC, including those that increase antioxidant activities in the bees’ hemolymph. This, in turn, contributed to the bees’ enhanced immunity, as the antioxidant system is the first line of humoral immunity activated after a pathogen breaches anatomical and physiological barriers [8,25].

Figure 10, in turn, presents the results of the measurements of the fluorescence anisotropy for the selected solvents and the water/sugar mixture. Measurements of anisotropy for aggregating systems tend to yield quite distinct changes. Thus, we can observe here an evident increase in the level of fluorescence anisotropy in an environment which is conducive to molecular aggregation. Depending on the solvent tested, the value of the fluorescence anisotropy varied from 0.02 (in MeOH) up to 0.22 in the water/sugar mixture.

Next, the fluorescence lifetimes for 7DOC were measured with the use of the TCSPC technique in several different solvents: acetonitrile, ethanol, isopropanol, butanol, THF, water and the sugar syrup used for bee treatment (Figure 11). The decays could be well fitted to two-exponential models, with the exception of the butanol solution, which required a third component (Table 3). The mean fluorescence lifetimes are within the nanosecond range for the organic solvents but decrease to the sub-nanomolar range for the water and glucose syrup. A polar microenvironment causes a decrease in both component lifetimes as well as an increase in the fractional intensity of the shorter component. This effect is further influenced by intermolecular aggregation, which—by quenching the fluorescence emission—increases the efficiency of the non-radiative channels of deactivation of the excited state of the 7DOC molecule.

## 3. Materials and Methods

### 3.1. Determination of the Biological Properties

#### 3.1.1. Experimental Protocols/Sampling

This study was performed at the apiary of the University of Life Sciences in Lublin, Poland (51.224039° N-22.634649° E).

The coumarin derivative used in the study (7DOC) was purchased from MERCK. Ten queenless colonies were prepared so that each contained nine Dadant (Dadant Blatt from Łysoń Beekeeping Company, Klecza Górna, Poland; 20 frames; 435 × 150 mm) combs plus a feeder containing sucrose syrup. Eight-day-old *A. mellifera carnica* queens originating from the same mother-queen were instrumentally inseminated with the semen of drones from the same colony. The inseminated queens were individually introduced into the queenless colonies. After one month, the five queens that laid the most eggs were caged for 24 h within a queen-excluder comb-cage containing two empty combs for egg laying. Next, the queens were uncaged and the cages containing the combs with eggs were left within the colonies for further worker brood rearing. On the 19th day of the apian development, the combs with the already sealed worker brood were transferred to incubators, in which they were maintained within individual chambers for 1-day-old workers to emerge. The emerging workers were captured and 25 randomly chosen individuals were immediately selected for hemolymph collection while the rest were placed in 120 wooden cages (12 × 12 × 4 cm; 50 bees per cage) which were divided into two groups (60 cages each). Each cage had a glass front screen as well as ventilation and feeding slots. The workers were fed with sugar syrup (1:1) ad libitum using inner feeders. In each group, food was replenished every other day during the experiment. In the first group, the syrup was supplemented with 7DOC (Figure 1) at a concentration of 200 µg/mL, whereas in the second group, the control, 7DOC was not added to the syrup. The dose of 7DOC administered was based on previously conducted pilot studies, which used the following concentrations of this compound: 50, 100, 150, 200, 250 and 300 µg/mL. In this experiment, the optimal dose was selected, as it resulted in the longest survival of the bees. The cages were kept in optimal conditions of 35 °C and 65% relative humidity. In each of the groups, 20 of the 60 cages were designated for longevity tests and the remaining 40 were used for biochemical assays.

#### 3.1.2. Longevity Test

Dead workers were counted and removed from each cage every two days (in total, 4000 workers were tested: 2 groups × 20 cages × 50 workers).

#### 3.1.3. Hemolymph Collection

In each group, hemolymph was taken from 25 bees at the age of 1, 7, 14, 21, 28 and 35 days. A glass capillary (20 µL; ‘end-to-end’ type; without anticoagulant; Medlab Products, Raszyn, Poland) was individually inserted between the third and fourth tergite of living workers to obtain fresh hemolymph, according to Łoś and Strachecka’s [77] method. Hemolymph volumes were separately measured in each capillary. Hemolymph from each bee was collected into one sterile Eppendorf tube containing 25 µL of ice-cooled 0.6% NaCl. The hemolymph solution was immediately frozen at −80 °C for further biochemical analyses.

#### 3.1.4. Biochemical Analyses

Total protein concentrations were assayed with the Lowry et al. [78] method modified by Schacterle and Pollack [79].

The antioxidant activities were measured as follows:Superoxide dismutase (SOD) activity was determined using a commercial Sigma-Aldrich (19,160) SOD Determination Kit (Poznań, Poland);Catalase (CAT) activity was determined using a Catalase Assay Kit (219265-1KIT) from Sigma-Aldrich (Poznań, Poland);Glutathione peroxidase (GPx) activity was determined using a Glutathione Peroxidase Assay Kit (Sigma Aldrich, Schnelldorf, Germany, no. MAK437);Glutathione S-Transferase (GST) activity was determined using a Glutathione-S-Transferase (GST) Assay Kit (Sigma Aldrich, Saint Louis, MO, USA, no. CS0410-1KT);Total antioxidant capacity (TAC) was determined using a Total Antioxidant Capacity Assay Kit (MAK187-1KT) from Sigma-Aldrich (Poznań, Poland).

All the antioxidant enzyme activities were calculated per 1 mg of total protein.

#### 3.1.5. Statistical Analysis

The results were analyzed statistically using Statistica formulas (TIBCO Software, Palo Alto, CA, USA), version 13.3 (2017) for Windows StatSoft Inc., Tulsa, OK, USA. The distribution of the data was analyzed with the Shapiro–Wilk test. To compare the antioxidant activities between workers of different ages from the two groups (supplemented and not supplemented with 7DOC), a mixed-model two-way ANOVA was used. If the difference between the groups was statistically significant, the ANOVA procedure was followed by multiple comparison testing using the post hoc Tukey HSD test with *p* = 0.05 as the level of significance.

#### 3.1.6. Absorption Spectroscopy (UV-Vis), Electronic Fluorescence Spectroscopy and Resonance Light Scattering (RLS)

The electronic absorption spectra were recorded using a Cary 300 Bio dual-beam UV-Vis spectrophotometer (Varian, Melbourne, Australia), which is equipped with a heated holder with a 6 × 6 cell Peltier block. The temperature was controlled using a thermocouple (Cary Series II from Varian) placed directly into the sample during measurement.

A Cary Eclipse spectrofluorometer (Varian, Melbourne, Australia) was used to measure the excitation, emission and synchronous spectra; all the spectra were measured at room temperature. The fluorescence spectra were recorded with a resolution of 0.5 nm after appropriate correction of the spectral characteristics of the lamp and photomultiplier.

Measurements of synchronous RLS (resonance light scattering) spectra were performed according to a previously described protocol with synchronous scanning of both the excitation and emission monochromators (without a gap between the excitation and emission wavelengths) and with a spectral resolution of 1.5 nm. Grams/AI 8.0 software from Thermo Electron Corporation (Waltham, MA, USA) was used to analyze all the recorded data.

#### 3.1.7. Anisotropy Measurements

Steady-state fluorescence anisotropy (r) was calculated from the polarized components of the fluorescence emission according to the following Equation (2):(2)$$r=\frac{I_{VV}-GI_{VH}}{I_{VV}+2GI_{VH}}$$
where *I_VH_* is the intensity of the vertically excited, horizontally observed emission; *I_VV_* is the intensity of the vertically excited, vertically observed emission; and *G* is the geometrical factor correcting the system’s polarization bias. A wire grid polarizer was used for the excitation to allow UV light transmission.

#### 3.1.8. Fluorescence Lifetime Measurements—Time-Correlated Single Photon Counting (TCSPC)

TCSPC measurements were performed using a FluoroCube fluorimeter (Horiba, Grabels, France). The samples were excited with a pulsed NanoLED diode at 372 nm (pulse duration of 150 ps) operated with 1 MHz repetition. To avoid pulse pile-up, the power of the pulses was adjusted to an appropriate level using a neutral gradient filter. The fluorescence emission was recorded using a picosecond detector TBX-04 (IBH, JobinYvon, Northampton, UK). DataStation and DAS6 software (JobinYvon, IBH, Northampton, UK) was used for data acquisition and signal analysis. All the fluorescence decays were measured in a 10 × 10 mm quartz cuvette, using an emitter cut-off filter with transmittance for wavelengths longer than 408 nm. The excitation profiles required for the deconvolution analysis were measured without the emitter filters, using a light-scattering cuvette. All the measurements were performed at a temperature of 22 °C. Each fluorescence decay was analyzed with a multiexponential model expressed by Equation (3):(3)$$I_{t}=\sum_{i}\alpha_{i}exp\left(-\frac{t}{\tau_{i}}\right)$$
where $\alpha_{i}$ and $\tau_{i}$ are the pre-exponential factor and the decay time of component *i*, respectively. Best-fit parameters were obtained through minimization of the reduced value and residual distribution of the experimental data. The fractional contribution of each decay time and the average lifetime of fluorescence decay were calculated based on the following Equations (4) and (5):(4)$$f_{i}=\frac{\alpha_{i}\tau_{i}}{\sum_{j}\alpha_{j}\tau_{j}}$$(5)$$\langle \tau \rangle =\sum_{i}f_{i}\tau_{i}$$

#### 3.1.9. Single-Crystal X-Ray Diffraction Measurements

To obtain 7DOC crystals, 2.67 mg of powder green compound was dissolved in 1 mL of isopropyl alcohol. The mixture was stirred for 45 min at 70 °C. The color changed to dark red after dissolving. After the color change, the magnetic stirring elements were removed from each vial and the heating was turned off. The mixture was cooled to room temperature and the caps were slightly unscrewed to allow the solvent to evaporate. Dark brown crystals appeared after a week.

For a selected monocrystal, single-crystal X-ray diffraction data was collected using a SuperNova diffractometer controlled by CrysAlis PRO software. The diffractometer was equipped with a Cu Kα microfocus X-ray source (λ = 1.54 Å, 50.0 kV and 0.8 mA) and a HyPix detector. Data collection was performed at 100 K. For data reduction, CrysAlis PRO 1.171.42.70a was used. The structure was solved by SHELXL [80] and refined in the Olex2-1.5 [81] program.

Energy calculations for the molecule interactions in the crystal were made in CrystalExplorer 17.5 [82], with electron density modeled at the B3LYP/6-31G(d, p) level of theory.

## 4. Conclusions

Our studies have confirmed that 7DOC is a compound that affects the stimulation of the antioxidant system, which is one of the elements of the immune system of bees. Strengthening/activating antioxidant enzymes in the hemolymph of bees provides protection against unfavorable environmental factors (e.g., pesticides, acaricides, heavy metals), as well as against pathogens. The effect of 7DOC was to extend the bees’ lifespan to 104 days, which is most likely related to changes in gene expression and the epigenome.

Furthermore, the spectroscopic assays complemented with crystallographic ones showed that, depending on the environment, we observed an increase in the processes connected with molecular aggregation. This tendency was particularly evident in the aquatic environment and in the bee food composed of water/sugar syrup. In these environments, we could notice distinctly strengthened bands on the longwave side of the electron absorption spectra, indicating the formation of specific aggregated structures, which was confirmed more clearly by the fluorescence excitation spectra and RLS spectra. Such aggregation most likely affects the absorption of 7DOC from the bees’ digestive tract into their hemolymph and then into the fat body, where proteins, including immune ones, are produced.

We observed a decrease in the fluorescence intensity in the fluorescence emission spectra and, on the other hand, a much weaker glow at excitation in the maxima of the bands associated with the aggregated forms of the analyzed molecule. These observations were confirmed by the measurements comprising fluorescence lifetimes, where the lifetime of a molecule in the excited state in the aquatic environment and in the water/sugar environment was considerably shorter. Furthermore, the crystallographic assays unquestionably confirmed the possible occurrence of a few aggregation structures, which can also form even more readily in environments of different solvents.

Finally, it needs to be added that the study presented in this article confirms that the analyzed compound is a promising choice to be used in apiculture, which in future stages of our research will be one of the application aims, although this will require studies in apiary conditions.

## Data Availability

The datasets used and/or analyzed during the current study are available from the corresponding author on reasonable request.

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
