# Peer review of "Advanced Research on Biological Properties—A Study on the Activity of the Apis mellifera Antioxidant System and the Crystallographic and Spectroscopic Properties of 7-Diethylamino-4-hydroxycoumarin"

_ijms, 2025, doi:10.3390/ijms26147015_

Round 1
Reviewer 1 Report
Comments and Suggestions for Authors
The introduction provides an excellent theoretical fundamentation.
The material and methods are very well detailed and can be replicated.
The statistics used to analyze the data are appropriate, given the distribution of the data.
The results presented were well discussed and very important for the improvement of the sector.
And the conclusions reflect the objectives.
Author Response
Report (Reviewer 1)
The introduction provides an excellent theoretical fundamentation.
The material and methods are very well detailed and can be replicated.
The statistics used to analyze the data are appropriate, given the distribution of the data.
The results presented were well discussed and very important for the improvement of the sector.
And the conclusions reflect the objectives.
Thank you for your effort in reviewing, as well as for appreciating our work and for your objective and kind review.

Reviewer 2 Report
Comments and Suggestions for Authors
Firstly, I would like to express that the article titled 'Advanced research on biological properties: a study on the activity of the Apis mellifera antioxidant system of the crystallographic and spectroscopic properties of 7-diethyloamino-4-hydroxycoumarin' is of great interest from both a scientific and a practical point of view.
Nevertheless, I have some questions and comments:
- What does the title ‘advanced research on biological properties’ refer to?The description suggests that the authors focused on determining protein concentrations and the activity of SOD, CAT, GPx, as well as GST and TAC levels. Can this research be considered ‘advanced’?
- The authors provide a thorough account of the spectroscopic and crystallographic analyses of 7-diethyloamino-4-hydroxycoumarin. However, I could find no justification for conducting these tests, nor any explanation of their connection with the biological activity of the coumarin under study and its possible practical application, in the abstract, discussion or conclusions. This issue should be discussed more clearly and comprehensively.
- The materials and methods section does not provide any information about the origin of the tested coumarins.
- With regard to lines 73–74 in the introduction, the extraction yield of coumarins in the extracts depends on the extraction techniques and the plant material extracted. Moreover, modern isolation techniques such as countercurrent chromatography allow coumarins to be obtained in sufficient quantities and of high enough purity for various biological tests.
Author Response
Report (Reviewer 2)
Firstly, I would like to express that the article titled 'Advanced research on biological properties: a study on the activity of the Apis mellifera antioxidant system of the crystallographic and spectroscopic properties of 7-diethyloamino-4-hydroxycoumarin' is of great interest from both a scientific and a practical point of view.
Thank you for your effort in reviewing, as well as for appreciating our work and for your objective and kind review.
Nevertheless, I have some questions and comments:
- What does the title ‘advanced research on biological properties’ refer to?The description suggests that the authors focused on determining protein concentrations and the activity of SOD, CAT, GPx, as well as GST and TAC levels. Can this research be considered ‘advanced’?
In this publication, we examined the biological potential of 7-Diethylamino-4-hydroxycoumarin. To this end, we conducted studies using a biological model, the honeybee, as well as crystallographic and spectroscopic studies of this compound. Considering the number of analyses presented in the paper, it can be concluded that the activity, properties, and biological potential of this compound are unquestionable, even advanced. Of course, considering only the antioxidant potential and antioxidant activity assays, it is difficult to speak of advanced research. In the title, we intended a comprehensive, interdisciplinary approach to demonstrating the properties of 7-Diethylamino-4-hydroxycoumarin. We believe that the word "advanced" can be retained in the title. Another word that could be used is "interdisciplinary," as this research combines strictly biological research with chemical and physical studies, as presented in our publication. The decision to use the terms "advanced" or "interdisciplinary" is left to the Editor.
- The authors provide a thorough account of the spectroscopic and crystallographic analyses of 7-diethyloamino-4-hydroxycoumarin. However, I could find no justification for conducting these tests, nor any explanation of their connection with the biological activity of the coumarin under study and its possible practical application, in the abstract, discussion or conclusions. This issue should be discussed more clearly and comprehensively.
Thank you for your comment. To enhance and further highlight the biological effects of 7-diethyloamino-4-hydroxycoumarin and link them to its chemical and physical properties, we have added /changed the following sentences:
In abstract
The fragment “Next, measurements of UV-Vis spectra, emission and excitation of fluorescence, synchronous spectra and finally of fluorescence lifetimes were made with the time-correlated single proton counting method, in various environments differing in polarity and in the environment applied in bee research.” has been replaced with “To confirm these potent biological properties of 7DOC, UV-Vis spectra, emission and excitation of fluorescence, synchronous spectra and finally of fluorescence lifetimes of this compound were measured using the time-correlated single proton counting method, in various environments differing in polarity and in the environment applied in bee research. This compound was shown to be sensitive to changes in solvent polarity.”
In discussion:
The following sentences were added:
“However, it can be assumed that such particle aggregation may promote the absorption of food/syrup containing 7DOC through the bee's digestive tract into its hemolymph and then into the fat body, where proteins and other compounds, including those included in the antioxidant system, are synthesized [46]. Hence, the high antioxidant activities observed in our studies, shown in Figs. 2-4.”
“These solvents are fundamental solutions in beekeeping. Water is essential to the lives of bees and all other creatures on Earth, and the water:sugar solution is used by beekeepers to feed colonies before wintering and during the nectarless periods. During these periods, biostimulators, protein products, amino acids, or pollen are also fed to bees in syrup or sugar candies [23-34]. Undoubtedly, 7DOC can be added to this list. The structure of this compound, based on two main aromatic rings: a benzene ring and a heterocyclic lactone ring, is compatible with naturally occurring compounds (e.g., in plant pollen), whose potential is utilized by bees [79-81]. 7DOC, as a coumarin derivative, similarly likely inhibits the proliferation of the bee sacbrood virus (CSBV) in the bee's body and reduces the number of copies of this virus; at the same time, it may induce the expression of an endogenous antibacterial peptide and improve the innate immunity of bees [82]. These compounds increase the survival of bee larvae [83] and imago, which was also confirmed by our research (Figure 1).”
“This likely influences the activity and biological properties of 7DOC, including those that increase antioxidant activities in the bees' hemolymph. This, in turn, contributed to the bees' enhanced immunity, as the antioxidant system is the first line of humoral immunity activated after a pathogen breaches anatomical and physiological barriers [8, 25].”
In conclusion:
The following sentences were added: “Such aggregation most likely affects the absorption of 7DOC from the bees' digestive tract into their hemolymph and then into the fat body, where proteins, including immune ones, are produced.”
- The materials and methods section does not provide any information about the origin of the tested coumarins.
We would like to sincerely thank the Reviewer for this very valid and important comment. We are not sure how this information escaped our attention, and we apologize for this oversight. Of course, the compound used in the final experiments was purchased from Merck, and we have now added this information to the Materials and Methods section.
- With regard to lines 73–74 in the introduction, the extraction yield of coumarins in the extracts depends on the extraction techniques and the plant material extracted. Moreover, modern isolation techniques such as countercurrent chromatography allow coumarins to be obtained in sufficient quantities and of high enough purity for various biological tests.
Thank you for your comment. We have added the information suggested by the reviewer to our manuscript.

Reviewer 3 Report
Comments and Suggestions for Authors
This manuscript presents an interdisciplinary study combining biochemistry, spectroscopy and apiculture. The work explores the effects of 7-diethylamino-4-hydroxycoumarin (7DOC) on the antioxidant system in honeybees and its physicochemical properties. The results are novel and potentially impactful for the field of bee health and apiculture.
However, the manuscript needs some improvements:
Line 118–121: Consider adding p-values ​​or confidence intervals to support this claim.
Line 135–155 and all figures explanations: ANOVA results are presented but overly technical. Consider including summary interpretation as well as stars of different letters where differences were significant so figures will be self-explanatory.
Line 174–176: While 104 days is biologically possible (especially for winter bees), this value is exceptionally high for caged summer bees (European Apis mellifera) under lab conditions (almost impossible). More clarity is needed regarding whether these were winter bees or artificially prolonged. At least find other scientific papers with similar findings. I have a lot of experience in cage experiments on bees and 20-30 days is mostly the lifespan of caged bees, up to 50 could be possible to with some special adaptations of food etc., but I never heard or read in a scientific paper that caged summer bees lived such a long time (not even for summer bee in the hive). I have serious doubts whether this is possible. This is my major concern regarding this manuscript.
Line 391: "...the syrup was supplemented with 7DOC... at the concentration of 200 µg/mL." Why did you choose this concentration? Was it based on prior studies, toxicity assays, or a dose-response curve? Include a reference or pilot data that informed this dosage.
Author Response
Report (Reviewer 3)
This manuscript presents an interdisciplinary study combining biochemistry, spectroscopy and apiculture. The work explores the effects of 7-diethylamino-4-hydroxycoumarin (7DOC) on the antioxidant system in honeybees and its physicochemical properties. The results are novel and potentially impactful for the field of bee health and apiculture.
Thank you for your effort in reviewing, as well as for appreciating our work and for your objective and kind review.
However, the manuscript needs some improvements:
Line 118–121: Consider adding p-values ​​or confidence intervals to support this claim.
Thank you for the reviewer's comment. All data were statistically significant, and we have added a p-value.
Line 135–155 and all figures explanations: ANOVA results are presented but overly technical. Consider including summary interpretation as well as stars of different letters where differences were significant so figures will be self-explanatory.
As suggested by the reviewer, we added lowercase letters to the figures to represent statistically significant differences between groups.
Line 174–176: While 104 days is biologically possible (especially for winter bees), this value is exceptionally high for caged summer bees (European Apis mellifera) under lab conditions (almost impossible). More clarity is needed regarding whether these were winter bees or artificially prolonged. At least find other scientific papers with similar findings. I have a lot of experience in cage experiments on bees and 20-30 days is mostly the lifespan of caged bees, up to 50 could be possible to with some special adaptations of food etc., but I never heard or read in a scientific paper that caged summer bees lived such a long time (not even for summer bee in the hive). I have serious doubts whether this is possible. This is my major concern regarding this manuscript.
Thank you for your comment. Like the reviewer, we are surprised by this result. We also have considerable experience with cage experiments on bees. Professor Strachecka is the mentor of the team conducting research on bees in this field. We have numerous publications in this field; here are a few examples:
- Changes in the activities of antioxidant enzymes in the fat body and hemolymph of Apis mellifera L. due to pollen monodiets. Antioxidants 2025, 14, 69. DOI: 10.3390/antiox14010069
- The effect of pollen monodiets on fat body morphology parameters and energy substrate levels in the fat body and hemolymph of Apis mellifera L. workers. Sci. Rep. (Nat. Publ. Group) 14 (15177), DOI: 10.1038/s41598-024-64598-0
- Cannabidiol (CBD) supports the honeybee worker organism by activating the antioxidant system. Antioxidants 12(2): 279, DOI: 10.3390/antiox12020279
- CBD supplementation has a positive effect on the activity of the proteolytic system and biochemical markers of honey bees (Apis mellifera) in the apiary. Animals 12(18): 2313, DOI: 10.3390/ani12182313
- Cannabis extract has a positive–immunostimulating effect through proteolytic system and metabolic compounds of honey bee (Apis mellifera) workers. Animals 11(8): 2190, DOI: 10.3390/ani11082190
- Piperine as a new natural supplement with beneficial effects on the life-span and defence system of honeybees. J. Agric. Sci. 157(2): 140-149, DOI: 10.1017/S0021859619000431
- Varroa treatment with bromfenvinphos markedly suppresses honeybee biochemical defence levels. Entomol. Exp. Appl. 160(1): 57-71, DOI: 10.1111/eea.12451
- Curcumin stimulates biochemical mechanisms of Apis mellifera resistance and extends the apian life-span. Journal of Apicultural Science 59(1): 129-141, DOI: 10.1515/jas -2015-0014
- Coenzyme Q10 treatments influence the lifespan and key biochemical resistance systems in the honeybee, Apis mellifera. Archives of Insect Biochemistry and Physiology, 86(3): 165-179; DOI: 10.1002/arch.21159
- Unexpectedly strong effect of caffeine on the vitality of western honeybees (Apis mellifera). Biochemistry (Moscow), 79(11): 1192-1201, DOI: 10.1134/S0006297914110066
We cannot explain this phenomenon. We conducted the experiment in the summer, beginning at the end of June. It is impossible that a generation of winter bees existed in Poland at that time. We also cannot find any publications demonstrating such long-lived bees. Even in our earlier experiments, using foods with biostimulators, we extended the lifespan of bees to approximately 50 days; for example, in the publication: DOI: 10.1134/S0006297914110066.
We can assume that 7DOC influenced gene expression and most likely the epigenome. However, this requires further research. Currently, the results are so satisfactory that we did not want to wait to publish them. To study gene expression/epigenome, we need to obtain funding/a project for this purpose.
Line 391: "...the syrup was supplemented with 7DOC... at the concentration of 200 µg/mL." Why did you choose this concentration? Was it based on prior studies, toxicity assays, or a dose-response curve? Include a reference or pilot data that informed this dosage.
In our pilot studies, we used several doses of 7DOC: 50; 100; 150; 200; 250; and 300 µg/ml. We did not present all the results, as the publication would have been very lengthy. We selected the optimal dose that had a stimulating effect and presented these results in our publication. We added information about selecting the optimal dose in the publication.

Round 2
Reviewer 3 Report
Comments and Suggestions for Authors
The authors answered all issues and made all necessary changes according to my comments and suggestions.
The only thing I still would like to be improved is adding one or few sentences at the end of discussion or in the conclusion section, with explanation for bee longevity of more than 100 days. As you describe in your answer to me, that this is surprisingly high for you as well and there is no similar data in literature, so further investigation will be conducted confirm this and possibly revealed the mechanisms.
Author Response
Report (Reviewer 3)
The authors answered all issues and made all necessary changes according to my comments and suggestions.
The only thing I still would like to be improved is adding one or few sentences at the end of discussion or in the conclusion section, with explanation for bee longevity of more than 100 days. As you describe in your answer to me, that this is surprisingly high for you as well and there is no similar data in literature, so further investigation will be conducted confirm this and possibly revealed the mechanisms.
Thank you for appreciating our corrections to improve the work. Following the reviewer's suggestion, we have added information in both the discussion and conclusion about the effect of 7DOC on bee lifespan, which was 104 days. We have also added information that there is no evidence in the literature that any compound in cage conditions extends bee lifespan by this much.
